# Pyrolysis and Co-Combustion of Semi-Dry Sewage Sludge and Bituminous Coal: Kinetics and Combustion Characteristics

**Guangyang Li** [1], **Zhuoyuan Chen** [1], **Afeng Wu** [2], **Tao Shi** [2], **Xiong Zhang** [1,*], **Hui Li** [3], **Haiping Yang** [1], **Jingai Shao** [1], **Shihong Zhang** [1] **and Hanping Chen** [1]

[1] State Key Laboratory of Coal Combustion, School of Energy and Power Engineering, Huazhong University of Science and Technology, 1037 Luoyu Road, Wuhan 430074, China

[2] China Energy Engineering Group Guangdong Electric Power Design Institute Co., Ltd., Guangzhou 510663, China

[3] State Key Laboratory of Utilization of Woody Oil Resource, Hunan Academy of Forestry, Changsha 410004, China

[*] Correspondence: zhangxiong107@163.com; Tel.: +86-27-87542417; Fax: +86-27-87545526

**Abstract:** To reduce the energy consumption and cost of the drying of sewage sludge (SS) and to ensure stability during combustion, the pyrolysis and co-combustion characteristics of semi-dry SS after the dehydration of flocculant and bituminous coal (BC) were studied in this work. The results show that the decrease in moisture content accelerates the release of volatile substances, and the increase in heating rate can also enhance the release of water and volatile matters. Furthermore, in the co-combustion of semi-dry SS and BC, the increase in mixing ratio (from 0% to 60%) of semi-dry SS caused the ignition and burnout temperature to decrease from 481 °C to 214 °C and from 702 °C to 627 °C, respectively. During co-combustion, the infrared spectra showed that the temperature range of 300–700 °C was the main gas precipitation area, and the main gaseous products were $CO_2$, $NO_x$, $SO_2$, and volatile organic pollutants (VOCs).

**Keywords:** pyrolysis; co-combustion; semi-dry; sewage sludge; characteristics





## 1. Introduction

SS reduction, harmlessness, stabilization, and resource recovery have attracted much attention. In 2018, sludge production from urban wastewater treatment plants in China reached 67.65 million tons [1]. It is expected that China's annual sludge output will exceed 80 million tons in 2022–2025. There are several technologies for utilizing sludge, such as land application [2], making building materials [3], pyrolysis [4], gasification [5], and combustion [6]. Combustion is an important method for sludge utilization, and has the advantages of large volume reduction of sludge, reduction in the pollution of harmful substances (such as various germs, parasites, and heavy metals) to the environment, and the effective recycling of the energy in the sludge. The combustion temperature range of sludge is generally from 200 °C to 600 °C, with a burnout temperature of about 800 °C [7,8]. The combustion of sludge can be carried out using existing mature coal-fired facilities, which is conducive to the large-scale reasonable disposal of sludge.

Combustion is a promising method of sewage sludge (SS) disposal, but SS cannot be disposed of directly for combustion. The moisture content of SS is high, which causes instability in the combustion, so there is a requirement for drying before the combustion and drying can increase the calorific value of SS for better combustion [9,10]. Hao et al. [11] constructed a pilot plant for SS combustion and found that integrated drying and combustion is a feasible method to dispose of SS. Liang et al. [12] found that low-moisture SS can increase the temperature of combustion and effectively improve the burnout rate of fuel. When the moisture content decreased from 80% to 20%, the boiler outlet temperature

increased by approximately 108 °C, exhibiting a considerable improvement in the combustion characteristics of SS. However, the dewatering technology used for the pre-treatment of SS combustion is consumes a large amount of energy, and the combustion of single SS is not effective [13,14].

It is necessary to find a way to improve the combustion characteristics of SS for reducing the energy consumption and cost due to the drying. Blending with a high-grade fuel, such as bituminous coal (BC), is an effective way to dispose of SS, which can enhance the combustion characteristics of SS. The moisture content of the original SS is 88–90%, which is not easy to transport for its high liquidity, so its dehydration treatment is generally carried out. Xu et al. [15] reduced the moisture content of SS to about 50%, and compared with original SS, the volatile content and fixed carbon were reduced and the ash content was increased, which was favorable for combustion. Chen et al. [16] studied the thermochemical and kinetic behavior of coal co-combustion with SS by thermodynamic analysis, and the result showed that the co-combustion facilitated the combustion and reduced the emissions of toxic elements. The moisture content is 47–70% after the dehydration with a flocculant [17], referred to as semi-dry SS, which is suitable for the blending with BC, considering the energy consumption and cost of the drying [18,19]. However, there are few reports on the co-combustion process and characteristics of semi-dry SS and BC, which is of great significance for its large-scale treatment [20,21].

Therefore, to reduce the energy consumption and cost in the drying of SS and ensure the stability during combustion, in this study, the pyrolysis and co-combustion processes and characteristics of semi-dry SS after the dehydration of flocculant and BC were studied. The effects of different heating rates and moisture contents on the pyrolysis of SS and the effects of different combustion ratios, moisture contents, and heating rates of BC and semi-dry SS on the co-combustion were evaluated.

## 2. Results and Discussion

### 2.1. Pyrolysis Characteristics of Semi-Dry SS

In the TG and DTG curves of SS (Figure 1a,b), three distinct stages were identified [22,23]. The initial weight loss of 43.17% is related to the loss of physically absorbed water molecules in temperatures lower than 200 °C. The second weight-loss stage occurs in the range of 200–600 °C, accounting for about 14.19% of the total SS weight loss. The maximum weight loss occurs at 286–299 °C, and the side shoulder appears at about 450 °C. It is generally believed that the decomposition and transformation of aliphatic compounds in SS mainly occurs below 300 °C [24], which are mainly transformed into water, noncondensable gas, and tar. Above 300 °C, carbohydrate and protein compounds are pyrolyzed and converted into small molecules through the breaking of peptide bonds. Because the pyrolysis reaction kinetics of these two types of compounds are different, two weight-loss peaks appear [25]. In the temperature range from 600 to 900 °C, SS shows a weight loss of 0.81% of its original mass, slowly approaching a final residue weight of 59.59%. This stage is marked by an approximately constant rate of weight loss. The residual organic matter continues to decompose and carbonize, and the final remaining products are fixed carbon and ash in the third stage.

In the volatilization releasing stage, with the decrease in moisture content, the trend of the water precipitation rate slows down. Water postpones the release of volatile matter and the decreases decomposition rate of organic components. In addition, with increasing heating rate, the weight-loss rate of SS gradually increases in the first stage. The high heating rate accelerates the reaction rate of the samples and reduces the burnout time. High heating rates benefit combustion mainly in the first and second stages. In the second stage, the maximum weight-loss rate first increases and then decreases (DTG [10 °C/min] < DTG [20 °C/min] < DTG [15 °C/min]), and the temperature corresponding to the maximum weight loss has a certain lag.

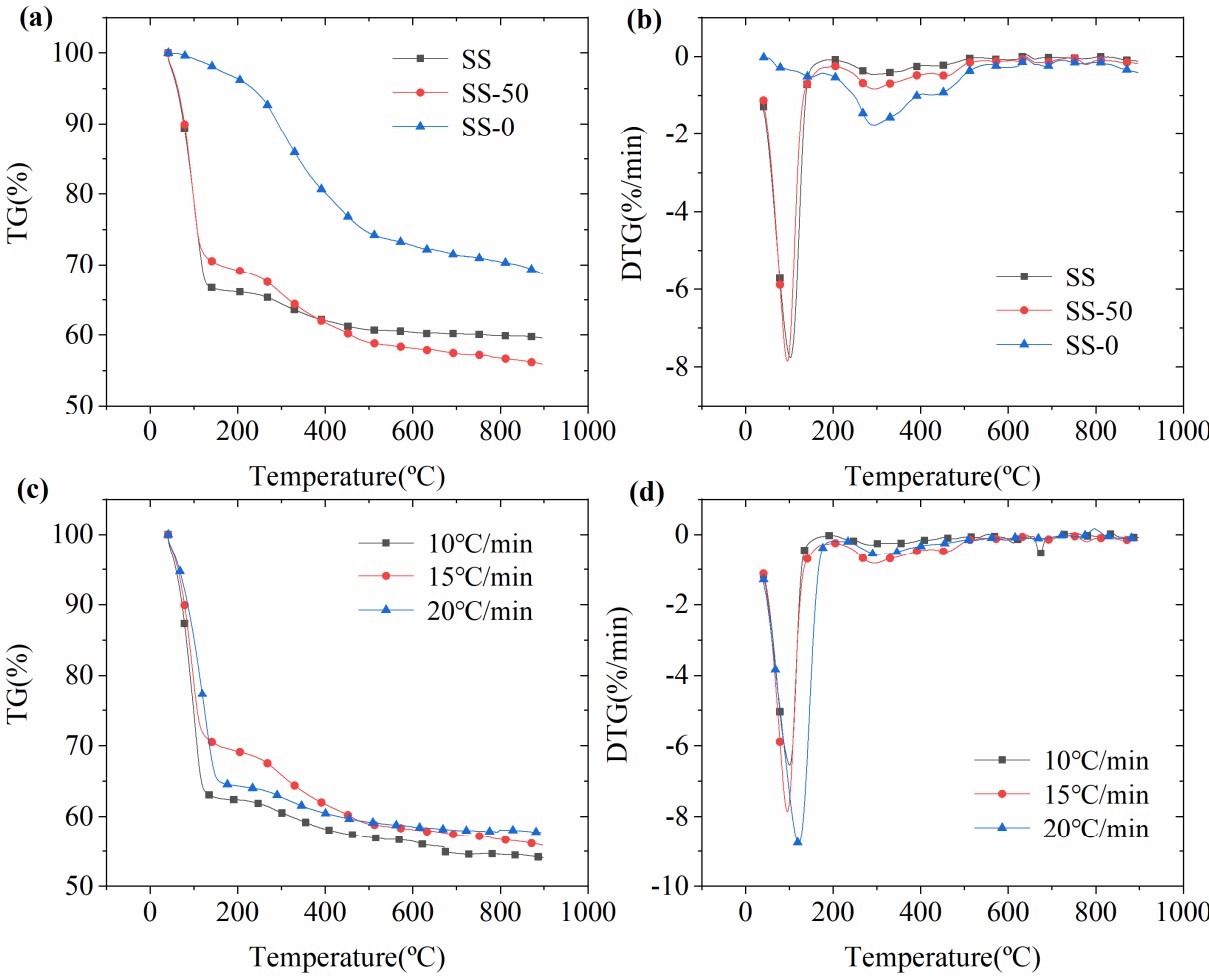

**Figure 1.** TG (**a**,**c**) and DTG (**b**,**d**) curves of SS pyrolysis at different moisture content and heating rates (SS-50).

As shown in Table 1, the maximum weight-loss rate of the water release phase increases from 6.55%·min$^{-1}$ to 8.78%·min$^{-1}$ with increasing heating rate, and the temperature corresponding to the maximum water release peak gradually increases from 101 °C to 121 °C. For the initiation stage of volatile release, with increasing heating rate, the overall trend of initial volatiles temperature gradually increases. The maximum weight loss first increases and then decreases. The temperature corresponding to the maximum volatiles releasing peak gradually increases. In addition, the TG curves shifted to the high-temperature side (right) as a whole. For semi-dry SS, the temperature corresponding to the maximum releasing peak of water and volatiles did not change much, but the maximum weight-loss peak and volatiles initial releasing temperature substantially changed. Owing to different heating rates, samples with higher heating rates experience a shorter reaction time and lower reaction degree, often resulting in thermal hysteresis. The higher the heating rate, the more obvious the thermal hysteresis effect, and the maximum weight-loss rate also increases, indicating that the pyrolysis reaction is more intense. As shown in Table 1, when the heating rate was 10, 15, and 20 °C/min, the final temperature residues were 54.1%, 55.9% and 57.8%, respectively, gradually increasing. For the pyrolysis process of SS, the D value first increases and then decreases with the heating rate. Among the three types of temperature, the optimum heating rate is 15 °C/min.

**Table 1.** Characteristics of the thermogravimetric curves of SS pyrolysis at different moisture content and heating rate.

| Samples | Heating Rate (°C/min) | $(dw_1/dt)_{max}$ /%·min$^{-1}$ | $T_{1max}$ /°C | $T_s$ /°C | $(dw_2/dt)_{max}$ /%·min$^{-1}$ | $T_{2max}$ /°C | $\Delta T_{1/2}$ /°C | R /% | $D \times 10^{-8}$ /%·min$^{-1}$·°C$^{-3}$ |
|---|---|---|---|---|---|---|---|---|---|
| SS-50 | 20 | 8.78 | 121 | 254 | 0.570 | 299 | 242 | 57.8 | 3.1013 |
| SS-50 | 10 | 6.55 | 101 | 216 | 0.298 | 286 | 182 | 54.1 | 2.6504 |
| SS-50 | 15 | 7.67 | 96 | 262 | 0.825 | 290 | 223 | 55.9 | 4.8734 |
| SS | 15 | 7.69 | 99 | 265 | 0.460 | 293 | 192 | 59.6 | 3.0856 |
| SS-0 | 15 | 0.52 | 149 | 225 | 1.760 | 295 | 137 | 68.7 | 19.3547 |

The TG and DTG curves of the samples with different moisture contents were substantially different. The maximum water releasing rate of the samples with higher moisture content is higher, and the maximum volatile release rate is lower. In addition, as shown in the variation of the D value from 3.0856 to 19.3547, with the decrease in the moisture content, the volatile release characteristic gradually enhanced.

### 2.2. Combustion of Semi-Dry SS and BC

#### 2.2.1. Single Combustion Characteristics of Semi-Dry SS

Figure 2 shows the combustion process of semi-dry SS at different heating rates and different moisture contents at a heating rate of 15 °C/min. As the heating rate increases from 10 °C/min to 20 °C/min, the overall DTG curves of SS move backward, and the ignition temperature gradually increases from 195 °C to 198 °C. Moreover, the burnout temperature also shifts backward, resulting in thermal hysteresis. The corresponding temperature of the maximum weight-loss peak of volatiles shifts forward, the corresponding maximum weight-loss rate increases, and the total weight-loss rate gradually decreases [26].

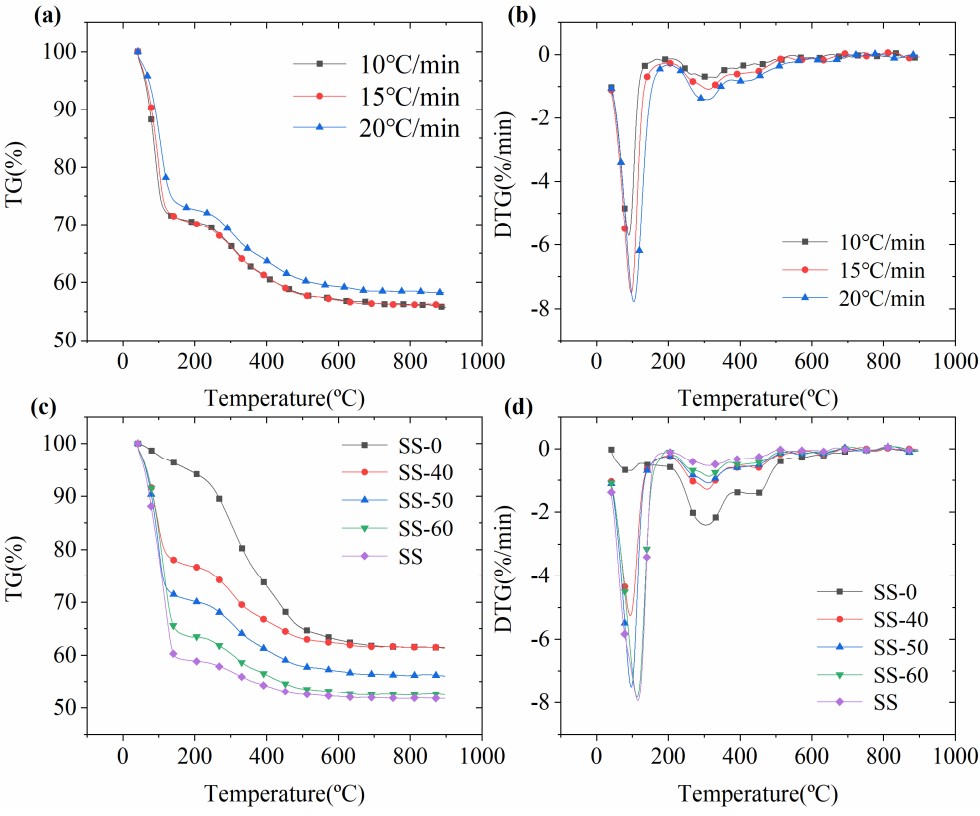

**Figure 2.** TG (**a,c**) and DTG (**b,d**) curves of SS pyrolysis at different heating rates (SS-50) and moisture contents.

With increasing moisture content, the total weight-loss rate of SS gradually increases, and the maximum peak value of the volatilization releasing peak gradually decreases. However, the temperature corresponding to the maximum peak value gradually increases from 303 °C to 312 °C. In the first stage, the release peaks of free water and bound water gradually increase, while in the second stage, the weight loss becomes more concentrated, the ignition temperature increases from 190 °C to 204 °C, and the burnout temperature decreases from 577 °C to 400 °C.

In Table 2, with the decrease in moisture content, the S value of SS increases from $1.16 \times 10^{-8}$ to $14.01 \times 10^{-8}$, the ignition temperature decreases from 204 °C to 190 °C and burnout temperature increases from 400 °C to 577 °C, and the intensity of maximum weight-loss peak increases from 0.53 to 2.39. Therefore, the more thorough the SS dewatering, the more favorable the combustion disposal, and the combustion characteristics of the SS-0 are better than those of BC. For SS and BC at different heating rates (10, 15, and 20 °C/min), the combustion characteristics show that the total mass-loss ratio of samples decreases with increasing heating rate. In addition, the DTG weight-loss curves of the sample have a high migration. The burning zone widens, and the peak and weight-loss rates of samples increase overall. Therefore, the C value and the S value increase. This is because some products are volatilized for a short time to produce a hysteresis phenomenon, and some combustible components need a higher temperature to escape, shifting the curves to a high-temperature region. However, with increasing heating rate, the reaction time becomes shorter, and the temperature difference between the inside and outside of the particles increases, affecting the internal combustion and making the reaction incomplete [27].

**Table 2.** Combustion characteristic parameters.

| Samples | $\beta$ (°C/min) | Characteristic Temperature/°C | | | DTG(%/°C) | | $C \times 10^{-6}$ | $G \times 10^{-6}$ | $S \times 10^{-8}$ |
| | | $T_i$ | $T_f$ | $T_{max}$ | $(dw/dt)_{max}$ | $(dw/dt)_{mean}$ | | | |
|---|---|---|---|---|---|---|---|---|---|
| SS-0 | 15 | 190 | 577 | 303 | 2.39 | 1.21 | 66.47 | 1154.59 | 14.01 |
| SS-40 | 15 | 194 | 486 | 307 | 1.28 | 0.69 | 34.05 | 571.71 | 4.84 |
| SS-50 | 15 | 196 | 478 | 312 | 1.08 | 0.63 | 28.17 | 460.66 | 3.44 |
| SS-60 | 15 | 198 | 444 | 312 | 0.87 | 0.53 | 22.36 | 361.95 | 2.68 |
| SS | 15 | 204 | 400 | 312 | 0.53 | 0.36 | 12.85 | 201.89 | 1.16 |
| SS-50 | 10 | 195 | 503 | 330 | 0.70 | 0.40 | 18.66 | 289.98 | 1.50 |
| SS-50 | 20 | 198 | 503 | 307 | 1.40 | 9.04 | 35.91 | 590.76 | 64.55 |

As shown in Table 3 and Figure 3, with increasing heating rate, the activation energy of the third stage of SS combustion decreases from 55.85 kJ/mol to 25.17 kJ/mol, which is consistent with the previous conclusion that the combustion process becomes more intense with at increased heating rates. When increasing the moisture content of SS, the activation energy of the third stage gradually decreases from 91.47 kJ/mol to 25.17 kJ/mol, and the activation energy of the second stage first decreases and then increases.

**Table 3.** Activation energy estimates based on the Coats–Redfern method.

| Samples | Heating Rate (°C/min) | Characteristic Temperature | | | | | | | | | | | |
| | | 75–240 °C | | | | 240–380 °C | | | | 380–600 °C | | | |
| | | Best-Fit Model | $A$ (s$^{-1}$) | $E$ (kJ/mol) | $R^2$ | Best-Fit Model | $A$ (s$^{-1}$) | $E$ (kJ/mol) | $R^2$ | Best-Fit Model | $A$ (s$^{-1}$) | $E$ (kJ/mol) | $R^2$ |
|---|---|---|---|---|---|---|---|---|---|---|---|---|---|
| SS-0 | 15 | 20 | $9.25 \times 10^2$ | 58.14 | 0.9493 | 9 | $4.73 \times 10^2$ | 62.07 | 0.9962 | 20 | $1.03 \times 10^7$ | 91.47 | 0.9788 |
| SS-40 | 15 | 20 | $1.30 \times 10^6$ | 55.21 | 0.7355 | 21 | 1.54 | 14.32 | 0.9761 | 20 | $3.09 \times 10^5$ | 62.74 | 0.9879 |
| SS-50 | 15 | 20 | 2.30 | 10.43 | 0.5827 | 20 | $3.59 \times 10^2$ | 28.80 | 0.9339 | 20 | $7.68 \times 10^5$ | 54.09 | 0.9953 |
| SS-60 | 15 | 20 | $1.88 \times 10^2$ | 21.11 | 0.5284 | 6 | $6.43 \times 10^{-3}$ | 2.98 | 0.7634 | 20 | $5.21 \times 10^5$ | 49.08 | 0.9933 |
| SS | 15 | 9 | $2.07 \times 10^{10}$ | 83.14 | 0.9947 | 20 | 1.67 | 4.91 | 0.6134 | 20 | $7.02 \times 10^2$ | 25.17 | 0.9851 |
| SS-50 | 10 | 9 | $4.76 \times 10^{13}$ | 105.22 | 0.9957 | 20 | 4.07 | 14.60 | 0.7967 | 20 | $3.95 \times 10^3$ | 55.85 | 0.9984 |
| SS-50 | 20 | 9 | $1.19 \times 10^{11}$ | 89.94 | 0.9932 | 20 | 0.18 | 4.26 | 0.6432 | 20 | $8.77 \times 10^3$ | 42.59 | 0.9931 |

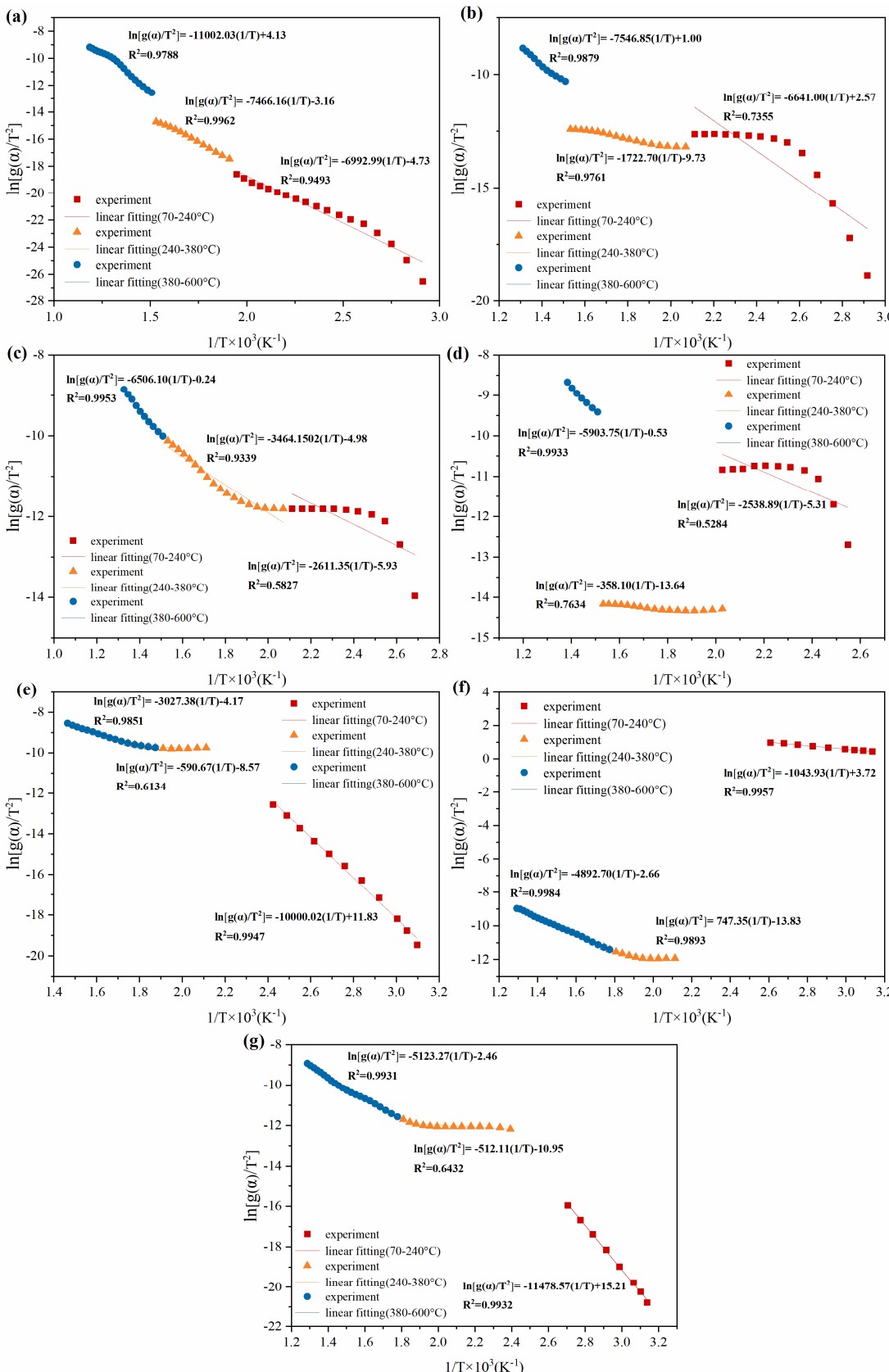

**Figure 3.** Linear fitting of the combustion kinetics of SS at different moisture contents and heating rates. (**a**) SS-0, 15 °C/min, (**b**) SS-40, 15 °C/min, (**c**) SS-50, 15 °C/min, (**d**) SS-60, 15 °C/min, (**e**) SS, 15 °C/min, (**f**) SS-50, 10 °C/min, (**g**) SS-50, 20 °C/min.

### 2.2.2. Co-Combustion of Semi-Dry SS and BC

Figure 4 shows the TG and DTG analysis results of the co-combustion of semi-dry SS and BC at different mixing ratios of semi-dry SS. In addition, the TG and DTG analysis results show that the BC and semi-dry SS have completely different combustion characteristics. The co-combustion of semi-dry SS and BC can be divided into three stages. The first stage is about the release of free water and bound water, because many hydrophilic groups are present in the organic matter of the SS, leading to a large weight-loss peak in this stage, and the temperature range of this stage is from 25 °C to about 200 °C. Owing to the low moisture content of BC, the weight loss is not obvious at this stage of the combustion of BC.

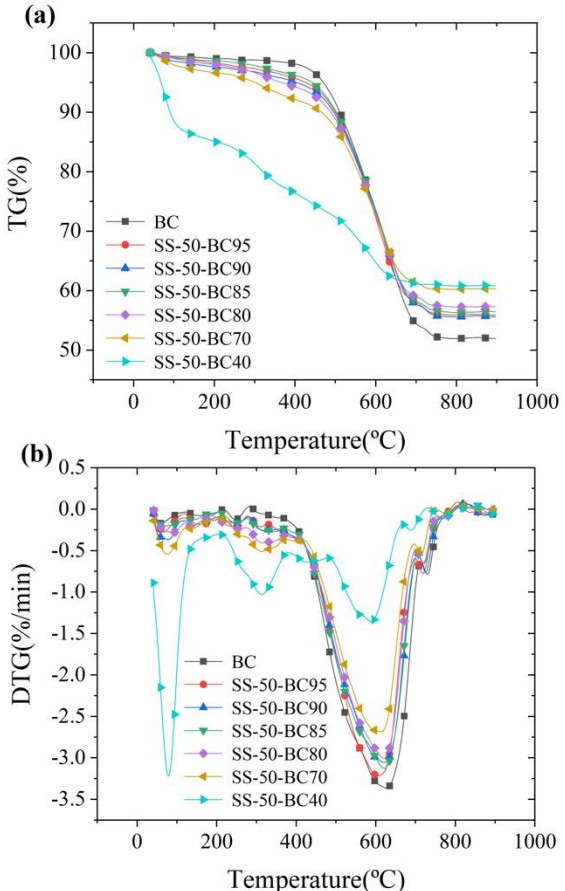

**Figure 4.** TG (**a**) and DTG (**b**) curves of semi-dry SS combustion at different mixed proportions.

The second stage involves the decomposition and combustion of semivolatile and volatile substances or organic compounds containing carbon. The temperature range is 200–450 °C. Generally, there are two more obvious peaks: at 300 °C, there is an obvious weight-loss peak, and at 420 °C, there is a side peak. The main weight-loss peak of DTG curves appears because a large amount of organic matter precipitates out. With gradually increasing temperature, hemicellulose and cellulose begin to decompose and burn to form secondary volatile peaks. The third stage involves the combustion of fixed carbon and residue, and the temperature range is mainly 450–800 °C. The weight loss of BC mainly occurs in this stage. The fixed carbon content of semi-dry SS is less, and the fixed carbon combustion peak is not obvious, which is fused with the secondary volatilization peak [28].

According to Figure 4, the TG curves shown the weight loss of BC was mainly concentrated on the combustion of fixed carbon in the third stage, while the weight loss of semi-dry SS was mainly concentrated in the first and second stages. When increasing the mixing ratio of semi-dry SS, the ignition temperature and burnout temperature gradually decrease. The maximum weight-loss peak decreases, and the peak position shifts [29]. Therefore, mixing semi-dry SS can improve the ignition characteristics of BC.

Table 4 shows that during the combustion of BC and semi-dry SS, when the semi-dry SS combustion content is less than 30%, the C value, the G value and the S value of semi-dry SS and BC are all lower than those of single BC. However, when the semi-dry SS mixing ratio increases to 60%, the C value and the G value of semi-dry SS combustion increases, but the S value is still lower than that of single BC combustion. This is mainly because the high moisture content clearly causes a decline in the largest weight-loss peak, ignition temperature and burning temperature. The ignition temperature and burnout temperature substantially decrease from 481 °C to 214 °C and from 702 °C to 627 °C, respectively, greatly improving the ignition characteristics. Therefore, the semi-dry SS is combustible and has a stable combustion index when increasing the mixing ratio of semi-dry SS. This may be because the semi-dry SS contains a variety of alkali metals and alkaline earth metals, such as Na, Ca, K, and Mg, which can catalyze and promote the reaction between free radicals and coal coke, thus improving the reactivity [30]. During the combustion process of the semi-dry SS, the pyrolysis gas produced by oxygen-rich substances can promote the heterogeneous gas–solid reaction of samples and enhance the combustion reaction [31].

**Table 4.** Combustion characteric parameters.

| Samples | $\beta$ (°C/min) | Characteristic Temperature (°C) | | | DTG (%/°C) | | $C \times 10^{-6}$ | $G \times 10^{-6}$ | $S \times 10^{-8}$ |
| | | $T_i$ | $T_f$ | $T_{max}$ | $(dw/dt)_{max}$ | $(dw/dt)_{mean}$ | | | |
| --- | --- | --- | --- | --- | --- | --- | --- | --- | --- |
| BC | 15 | 481 | 702 | 624 | 3.42 | 2.68 | 14.79 | 49.28 | 5.65 |
| SS-50-BC95 | 15 | 478 | 697 | 602 | 3.29 | 2.33 | 14.43 | 50.15 | 4.83 |
| SS-50-BC90 | 15 | 469 | 696 | 621 | 3.12 | 2.28 | 14.20 | 48.76 | 4.65 |
| SS-50-BC85 | 15 | 475 | 690 | 621 | 3.12 | 2.38 | 13.86 | 46.99 | 4.78 |
| SS-50-BC80 | 15 | 470 | 686 | 620 | 3.00 | 2.21 | 13.61 | 46.71 | 4.39 |
| SS-50-BC70 | 15 | 451 | 672 | 614 | 2.68 | 1.92 | 13.19 | 47.63 | 3.79 |
| SS-50-BC40 | 15 | 214 | 627 | 588 | 1.35 | 0.80 | 29.57 | 235.00 | 3.78 |

The co-combustion of semi-dry SS and BC can be regarded as a series of complex volatile releases and combustions. The consecutive Redfern dynamic method was used for the analysis. The combustion kinetic parameters of semi-dry SS with different mixing ratios are shown in Table 5.

**Table 5.** Activation energy estimates based on the Coats–Redfern method.

| Samples | Characteristic Temperature | | | | | | | | | | | |
| | 45–100 °C | | | | 100–380 °C | | | | 380–800 °C | | | |
| | Best-Fit Model | $A$ (s$^{-1}$) | $E$ (kJ/mol) | $R^2$ | Best-Fit Model | $A$ (s$^{-1}$) | $E$ (kJ/mol) | $R^2$ | Best-Fit Model | $A$ (s$^{-1}$) | $E$ (kJ/mol) | $R^2$ |
| --- | --- | --- | --- | --- | --- | --- | --- | --- | --- | --- | --- | --- |
| BC | 9 | $6.09 \times 10^5$ | 66.69 | 0.8361 | 20 | $2.22 \times 10^5$ | 20.16 | 0.9635 | 16 | $1.61 \times 10^3$ | 67.92 | 0.9958 |
| SS-50-BC95 | 9 | $1.18 \times 10^9$ | 86.20 | 0.8668 | 20 | $2.03 \times 10^{-3}$ | 35.30 | 0.9937 | 16 | $2.79 \times 10^2$ | 55.05 | 0.9772 |
| SS-50-BC90 | 9 | $5.33 \times 10^9$ | 88.27 | 0.8515 | 20 | $1.48 \times 10^{-3}$ | 24.31 | 0.9756 | 16 | $2.16 \times 10^2$ | 53.32 | 0.9613 |
| SS-50-BC85 | 9 | $1.50 \times 10^{10}$ | 120.09 | 0.7998 | 20 | $2.80 \times 10^{-3}$ | 33.48 | 0.9346 | 16 | $3.82 \times 10^2$ | 52.19 | 0.9744 |
| SS-50-BC80 | 9 | $2.06 \times 10^{12}$ | 137.99 | 0.8300 | 20 | $1.05 \times 10^{-2}$ | 39.10 | 0.9610 | 16 | $1.37 \times 10^2$ | 49.94 | 0.9566 |
| SS-50-BC70 | 9 | $6.54 \times 10^8$ | 79.85 | 0.9406 | 20 | $9.48 \times 10^{-2}$ | 28.83 | 0.9173 | 16 | 5.06 | 42.82 | 0.9102 |
| SS-50-BC40 | 9 | $2.81 \times 10^8$ | 84.65 | 0.9660 | 20 | $5.66 \times 10^{-1}$ | 16.52 | 0.8406 | 18 | $8.28 \times 10^2$ | 41.53 | 0.9158 |

According to the DTG curves of the co-combustion of BC and semi-dry SS and the linear fitting curves shown in Figure 5, the co-combustion can be divided into three stages, but the BC only has an obvious mass-loss peak in the third stage, and the temperature ranges of the three stages are 45–100, 100–380, and 380–800 °C, respectively. Activation energy has been the focus of intensive research as a potential that provides important data on the minimum energy required to support the reaction. Conforming to the Avrami–Erofeev equation, it can be observed that the activation energy of the reaction does not depend much on the temperature in a certain temperature interval, and can be obtained with relative accuracy. The main suitable reaction mechanism functions are the Z-L-T equation of 3D diffusion, Mample's one-line rule of random nucleation growth and the Avrami–Erofeev equation (shown in Table 5) [32].

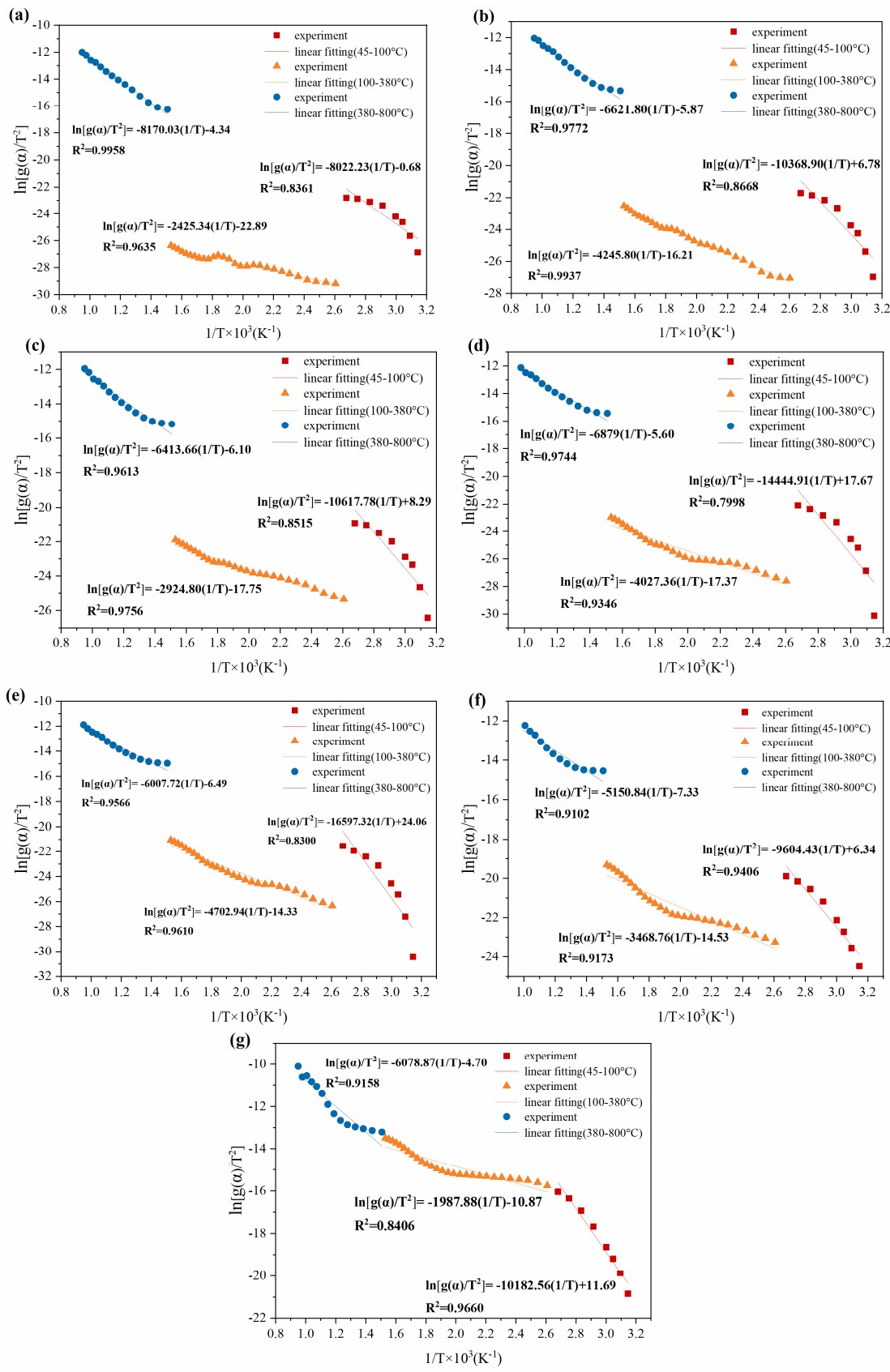

**Figure 5.** Linear fitting in combustion kinetics of semi-dry SS with different mixing ratio. (**a**) BC, (**b**) SS-50-BC95, (**c**) SS-50-BC90, (**d**) SS-50-BC85, (**e**) SS-50-BC80, (**f**) SS-50-BC70, (**g**) SS-50-BC40.

The first stage of semi-dry SS and BC mixing combustion conforms to the Z-L-T equation of 3D diffusion, but its linear regression correlation coefficient is between 0.83 and 0.94, meaning that the correlation is not high. This stage mainly involves water precipitation. The second stage conforms to the Mample single-line rule of random nucleation and the Avrami–Erofeev equation, and the combustion rate of blending in this stage is mainly affected by SS [33]. Moreover, the combustion in this stage belongs to the combustion of light components, and the activation energy required in this stage is less than that in the third stage. The activation energy of the third stage gradually decreases when the mixing ratio of semi-dry SS is increased. This is probably because semi-dry SS has a synergistic promoting effect on BC combustion, and alkali metal elements such as Na and K in the semi-dry SS have a catalytic effect on coal combustion, reducing the activation energy required for BC combustion [34].

The reaction kinetics models of semi-dry SS and BC are different. In addition, when the semi-dry SS is added, the activation energy increases in the first phase and second, and in the third phase, the activation energy decreases. Thus, it can be observed that semi-dry SS is a suitable clean fuel, but we also need to pay attention to adding a suitable proportion of the semi-dry SS to improve the BC combustion reaction kinetics.

### 2.3. Emission Characteristics of Gas Pollutants during Co-Combustion

The types of bonds and typical wavenumber values in the FTIR spectra are shown in Table S2 [35–37]. The 3-D profiles of FTIR spectra (Figures S1–S7) and the FTIR spectra (Figure 6) show that the gas emission mainly occurred between 300 and 700 °C, which is in agreement with the thermogravimetric data (Figure 4). Distinct absorbance peaks are displayed in the stage of 2250–2400 $cm^{-1}$, which is representative of $CO_2$ due to its indicative asymmetric stretching of the carbonyl group (C=O). In addition, the intensity of the peak first increases and then decreases. Moreover, a weak absorption peak (2100–2250 $cm^{-1}$) appears at about 450 °C (about 314 °C for SS-50-BC40), corresponding to the characteristic peak of CO. The release of CO is probably caused by the decomposition of decarboxylation and other organic compounds in SS cracking such as phenol, aldehyde and alcohol, which contain oxygen functional groups [38]. In addition, this is probably because coke reacts with $CO_2$ to produce CO with increasing temperature for the combustion of single BC. At a higher temperature (above 200 °C), a weak and wide absorption peak appears at 3500–4000 $cm^{-1}$, corresponding to the stretching vibration band of -OH, which can be attributed to the cracking of alcohols, phenols, and other substances [39].

It can be seen from Figure 6 that mixing semi-dry SS into BC slightly affects the gaseous reaction products at a low proportion. A strong $CO_2$ absorption peak appears at 2350 $cm^{-1}$ in the co-combustion of semi-dry SS and BC, and $CO_2$ is present throughout the entire reaction process. Moreover, when increasing the semi-dry SS mixing ratio, the characteristic peak of $CO_2$ (2250–2500 $cm^{-1}$) gradually shifts to a low temperature. For single BC combustion, a hydroxyl absorption peak (3450–4000 $cm^{-1}$) appears, mainly at 500–700 °C, during combustion. When the proportion of semi-dry SS is greater than 10%, a hydroxyl peak appears throughout almost the entire combustion process, indicating that when the proportion of semi-dry SS is high, the high water content substantially affects the substances containing -OH (including water) in the combustion. The absorbance bands of C-H (690–900 $cm^{-1}$) appearing at about 450 °C indicate aromatic evolution. The emission of $NH_4$ is confirmed by the appearance of absorption peaks (1550–1640 $cm^{-1}$) corresponding to N-H stretching vibration produced by the cleavage of nitrogen heterocyclic rings and the number of hydrogen-free radicals. The appearance of bands between 1680 and 1800 $cm^{-1}$ confirm the presence of carbonyl groups (aldehydes) in the evolved gases during the combustion of semi-dry SS and BC [40]. The reason for the formation of $NO_2$ (1497 $cm^{-1}$) and NO (931 and 965 $cm^{-1}$) might be because of the reaction of amino acids and other N-containing compounds in the SS with oxygen in the air [41]. The absorbance bands of S=O (1033 and 1054 $cm^{-1}$) appear, indicating the emission of $SO_2$, which might have originated from sulfur-containing substances. According to the above analysis, semi-dry

SS and BC mixed combustion produces gaseous substances including water, $CO_2$, CO, NO, $NO_2$, $SO_2$, $CH_4$, aromatic compounds, alkanes, olefins, aldehydes, and ketones.

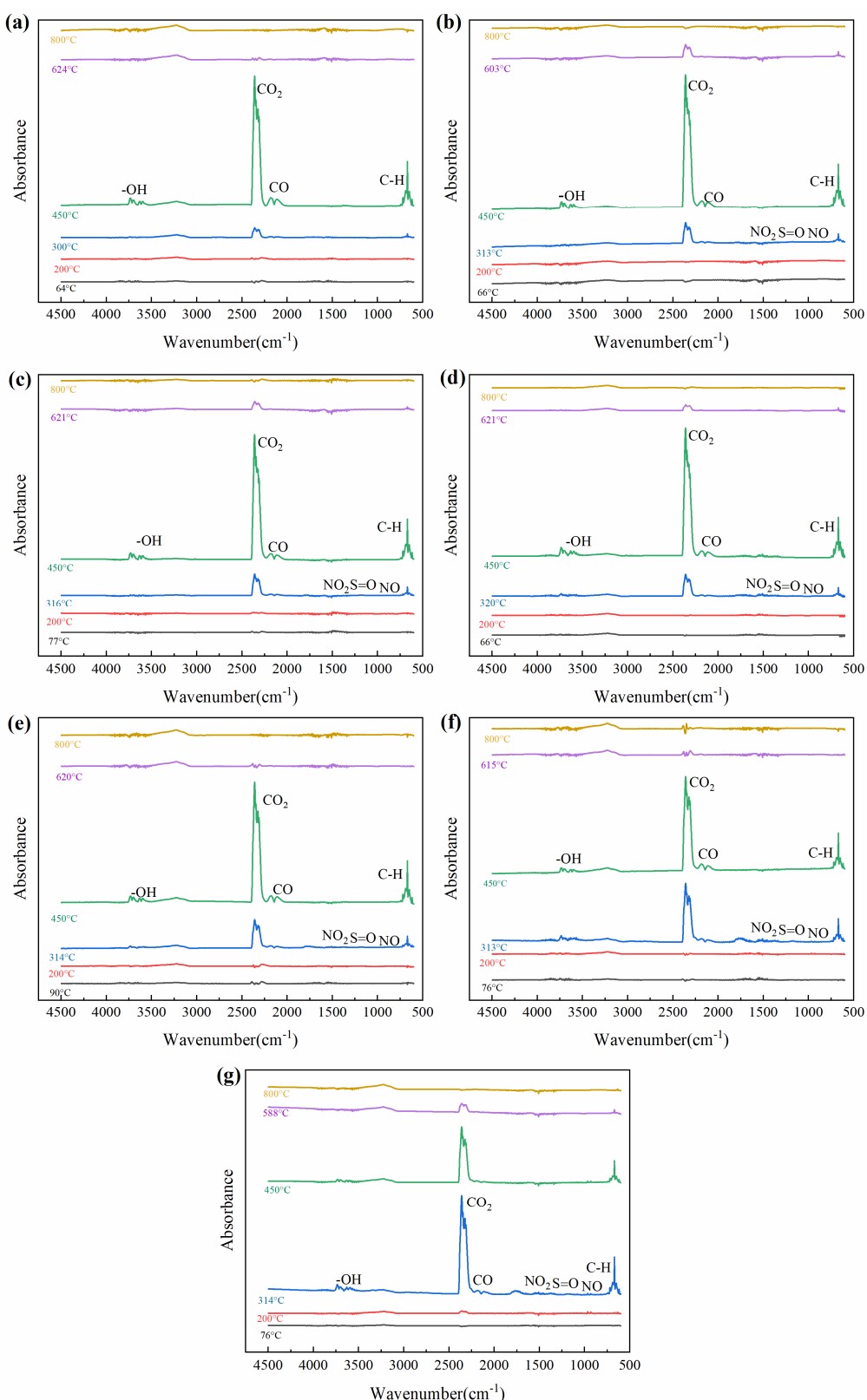

**Figure 6.** FTIR spectra for volatiles with different temperatures during the combustion process. (**a**) BC, (**b**) SS-50-BC95, (**c**) SS-50-BC90, (**d**) SS-50-BC85, (**e**) SS-50-BC80, (**f**) SS-50-BC70, (**g**) SS-50-BC40.

## 3. Materials and Methods

### 3.1. Materials

The SS sample was obtained from a sludge combustion plant in Zhongxiang of the Hubei province of China, and the BC was a kind of commercial product from the China Shenhua Shendong company. The samples were packed tightly in plastic containers and kept in a freezer at $-16$ °C. The SS sample was dewatered by adding ferric chloride as a flocculant. Proximate analysis was determined according to a proximate analysis instrument (Thermostep, Eltra, Germany) and elemental analysis of carbon (C), hydrogen (H), nitrogen (N) and sulphur (S) was carried out in an elemental analyzer (Vario Micro cube, Elementar, Germany). The calorific value of the samples was determined using an automatic calorimeter (Ika C6000, Ika, Germany). The proximate analysis, elemental analysis, and calorific values of SS and BC are shown in Table 6.

**Table 6.** Properties of the samples.

| Samples | Proximate Analysis/wt% | | | | Ultimate Analysis/wt% | | | | | HHV (kJ/kg) |
|---------|--------|--------|--------|--------|--------|--------|--------|--------|--------|--------|
| | $M_{ar}$ | $V_{ar}$ | $A_{ar}$ | $FC_{ar}$ | $C_{ar}$ | $H_{ar}$ | $N_{ar}$ | $S_{ar}$ | $O_{ar}$ * | $Q_{net,ar}$ |
| SS | 88.01 | 3.76 | 7.23 | 1.00 | 2.23 | 0.39 | 0.37 | 0.06 | 1.71 | 885.36 |
| BC | 6.71 | 32.88 | 19.27 | 41.14 | 61.76 | 4.16 | 1.11 | 0.52 | 6.47 | 23526.5 |

M: moisture; V: volatile matter; A: ash; FC: fixed carbon; O * content was calculated using the subtraction method; HHV: high heating value.

### 3.2. Thermogravimetric Analysis

The pyrolysis and combustion experiments were conducted using a TG analyzer (Evo1150, Setaram, France). The flow rate of $N_2$ (purity 99.999%) was 100 mL/min for the pyrolysis of SS and the air flow rate was 100 mL/min for the combustion of SS. The SS sample was dried in an oven at 105 °C to reduce its moisture content to 60%, 50%, 40% and 0%, which were coded as SS-60, SS-50, SS-40 and SS-0, and the sample (particle size 0.15–0.18 mm) mass was $10 \pm 0.1$ mg. To evaluate the effect of the heating rate, the temperature was increased from 25 °C to 900 °C at a heating rate of 10 °C/min, 15 °C/min, and 20 °C/min. A blank experiment was conducted following each condition to obtain the baseline to reduce the systematic errors. The TG analyzer was rapidly cooled down to complete the TG experiments after pyrolysis. To comprehensively evaluate the pyrolysis and combustion characteristics of SS, the TG curves and data were analyzed, and some evaluation parameters were used, including maximum precipitation rate $(dw_1/dt)_{max}$, maximum precipitation temperature $T_{1max}$, initial volatile temperature $T_s$, maximum volatile release rate $(dw_2/dt)_{max}$, $\Delta T_{1/2}$, $T_{2max}$, final temperature residue ratio R and volatile matter release characteristic index D, with more details shown in the Supplementary Materials.

### 3.3. Thermogravimetry–Fourier Transform Infrared Spectroscopy Analysis

To assess the co-combustion process and characteristics of semi-dry SS and BC and the evolution of gaseous products, the thermogravimetric analyzer (Sta 449F3, Netzsch, Germany) coupled to an FTIR spectrometer (Vertex80v, Bruker, Germany) was introduced to analyze the co-combustion with different mixing ratios of semi-dry SS. The air flow rate was 100 mL/min, and the combustion temperature range was 25–900 °C. For the co-combustion, semi-dry SS was blended with BC at the seven mass ratios of 0:1, 1:19, 1:9, 3:17, 1:4, 3:7 and 2:3, which were coded as BC, SS-50-BC-95, SS-50-BC-90, SS-50-BC-85, SS-50-BC-80, SS-50-BC-70 and SS-50-BC-40, respectively. To reduce the heat and mass transfer resistance, the single sample mass was set as $(10 \pm 0.5)$ mg. The heating rate was set as 15 °C/min. The transfer line and gas cell were heated at a temperature of 180 °C to avoid the condensation of volatile decomposition products. The 4 $cm^{-1}$ scan resolution was set in a wavenumber stage of 4500–600 $cm^{-1}$.

### 3.4. Combustion Index

To evaluate the combustion characteristics, the thermogravimetric curves and data were analyzed and processed, and adopted the evaluation parameters including ignition temperature $T_i$, burnout temperature $T_f$, corresponding temperature $T_{max}$, flammability index C, stable combustion index G and comprehensive combustion characteristic index S, with more details shown in Supplementary Materials.

### 3.5. Kinetic Analyses

The pyrolysis and combustion kinetics were described by the first-order Arrhenius law. The Coats–Redfern approximation expressed in logarithmic form is $\ln [G(\alpha)/T^2] = \ln [AR/E] - E/RT$ [42–44]. The expressions used for $G(\alpha)$ that can be used in the kinetic modeling to determine the mechanism involved are given in Table S1. The kinetic parameters E and A were estimated from the intercept and slope of the plot $\ln [G(\alpha)/T^2]$ vs. $1/T$. In addition, the $G(\alpha)$ expression having the highest correlation coefficient ($R^2$) was considered to be the best fitting, and was used for the kinetic modeling equation.

## 4. Conclusions

During the pyrolysis process of SS, with decreasing moisture content, the D value increased from $3.0856 \times 10^{-8}$ to $19.3547 \times 10^{-8}$, showing that the volatile release characteristic gradually increased. In addition, the higher the heating rate, the more obvious the thermal hysteresis in the water precipitation stage and the volatile release stage. At the same time, the maximum weight-loss rate also increased from $6.75\% \cdot min^{-1}$ to $8.78\% \cdot min^{-1}$, indicating that the pyrolysis reaction was more intense.

During co-combustion, the higher the moisture content of semi-dry SS, the lower the ignition temperature and the lower the burnout temperature. With increasing mixing ratio of semi-dry SS, the activation energy of the third stage gradually decreased, showing that the semi-dry SS had a synergistic effect on BC combustion. The FTIR results showed that 300–700 °C was the main gas releasing area during the second stage and the third stage of semi-dry SS and BC mixed combustion, and the combustion products included $CO_2$, CO, NO, $NO_2$, $SO_2$, and VOCs. Owing to the difference in the basic characteristics of semi-dry SS and BC, the ignition and burnout temperature decreased from 481 °C to 214 °C and from 702 °C to 627 °C, respectively, with increasing mixing ratio of semi-dry SS. The decrease in the moisture content and the increase in heating rate can obviously improve the C value, the G value, and the S value. When semi-dry SS is mixed with BC, BC combustion can improve the S value of SS, and SS combustion can improve the C value and the G value of BC.

**Supplementary Materials:** The following supporting information can be downloaded at: https://www.mdpi.com/article/10.3390/catal12101082/s1, Figure S1: 3-Dprofiles of FTIR spectra of evolved gases for volatiles with different temperatures during combustion process of BC; Figure S2: 3-Dprofiles of FTIR spectra of evolved gases for volatiles with different temperatures during combustion process of SS-50-BC95; Figure S3: 3-Dprofiles of FTIR spectra of evolved gases for volatiles with different temperatures during combustion process of SS-50-BC90; Figure S4: 3-Dprofiles of FTIR spectra of evolved gases for volatiles with different temperatures during combustion process of SS-50-BC85; Figure S5: 3-Dprofiles of FTIR spectra of evolved gases for volatiles with different temperatures during combustion process of SS-50-BC80; Figure S6: 3-Dprofiles of FTIR spectra of evolved gases for volatiles with different temperatures during combustion process of SS-50-BC70; Figure S7: 3-Dprofiles of FTIR spectra of evolved gases for volatiles with different temperatures during combustion process of SS-50-BC40; Figure S8: TG analyzer (Evo1150, Setaram, France); Figure S9: The thermogravimetric analyzer (Sta 449F3, Netzsch, Germany) coupled to an FTIR spectrometer (Vertex80v, Bruker, Germany); Table S1: Common reaction mechanism functions; Table S2: Types of bonds and typical wavenumber values during the Fourier transform infrared (FTIR) spectrometer process. References [45–54] are cited in the Supplementary Materials.

**Author Contributions:** G.L.: Conceptualization, Methodology, Investigation, Visualization, Data curation, Formal analysis, Writing—original draft. Z.C.: Writing—review and editing. A.W.: Writing—review and editing. T.S.: Writing—review and editing. X.Z.: Resources, Writing—review and editing, Visualization, Project administration, Funding acquisition. H.L.: Writing—review and editing. H.Y.: Writing—review and editing. J.S.: Writing—review and editing. S.Z.: Writing—review and editing, Supervision, Funding acquisition. H.C.: Writing—review and editing, Supervision, Funding acquisition. All authors have read and agreed to the published version of the manuscript.

**Funding:** This work was financially supported by the Natural Science Foundation of China (No. 52176187 and 51806077).

**Data Availability Statement:** Not applicable.

**Acknowledgments:** The authors wish to express the great appreciation of the financial support from the Natural Science Foundation of China (No. 52176187 and 51806077). The study was also supported by the Foundation of State Key Laboratory of Coal Combustion and benefits from the technical support from the Analytical and Testing Center in Huazhong University of Science and Technology (http://atc.hust.edu.cn).

**Conflicts of Interest:** The authors declare that they have no known competing financial interests or personal relationships that could have appeared to influence the work reported in this paper.

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
