# Peer review of "Pyrolysis and Co-Combustion of Semi-Dry Sewage Sludge and Bituminous Coal: Kinetics and Combustion Characteristics"

_catalysts, doi:10.3390/catal12101082_

Round 1

Reviewer 1 Report

This works studies the decomposition in the presence (or not) of O2. This is a very important work to be considered. In order to be improved and to be published, I recommend to do these alterations and comments:

1) Put the meaning of the acronyms before defining them.  

2) Is the combustion of any raw a sustainable alternative nowadays? Is it not the world seeing combustion as a "bad" technology? 

3) I suggest putting Table 1 in the Results and Discussion section.

4) The composition of the products of the combustion was poorly discussed. It is necessary to point out the disadvantages of this process in relation to the products.    

5) Why did you not keep the temperature constant (for instance, after the evaporation of water or after the release of the first species of waste) for a certain time? I recommend performing that and discuss. 

6) The English needs to be improved in some sections. 

Reviewer 2 Report

In this manuscript, the authors reported the pyrolysis and co-combustion of semi-dry sewage sludge and bituminous coal: thermodynamics, kinetics and combustion characteristics. In general, the reported works are interesting and could contribute to the significant added value especially the scientific knowledge related to energy generation. The manuscript is suggested to be accepted after the consideration of my specific comments as mentioned in the details below:

1. This manuscript only investigated the kinetic and missing info on thermodynamics such as enthalpy, Gibbs energy and entropy. Hence, the title must be revised by removing the keyword ‘thermodynamic’.

Introduction:

1. Line 33-37, The authors have clearly explained the combustion of sewage. However, lacking in describing the combustion process for example, temperature range, pressure range, and open or closed combustion. This information must be added in the introduction.

2. Line 52: State full name before abbreviation for BC.

3. Explained why semi-dry SS is considered compared to 100% dried SS.

4. Novelty of this work is not clear, revise again to highlight the novelty and research gap. The authors have highlighted lines 57 – 59 but not sufficient.

5. The authors stated that the combustion characteristics can be improved by blending SS and BC. Hence, why pyrolysis is needed? The objective and advantage of pyrolysis in this study are missing.

6. It's quite confusing, which one is affecting the combustion characteristics of SS?, is it drying of SS or blending with BC? If both means, then, the authors must revise lines 50-59. For example, Line 51-52, the authors stated that the blending method is good for high combustion characteristics of SS, while Line 55-56, the authors stated that 47-70% of moisture content is acceptable, so does with mean that SS undergoes drying first then blend with BC?

Methodology:

1. Line 104-105: On what basis the semi-dry SS to BC mass ratios were set? The increment in mass ratios is inconsistent, and justify in this setting.

2. Line 83-84: The authors stated that pyrolysis was conducted under nitrogen and combustion under air, hence, how does this work when the temperature is set at 25 – 900 C, at which temperature the combustion will occur? More explanation is needed on this.

3. Methodology on thermodynamic is missing. This manuscript only investigated the kinetic and missing info on thermodynamics such as enthalpy, gibbs energy and entropy. Hence, the title must be revised by removing the keyword ‘thermodynamic’.

Results & Discussion

1. Section 3.2.1: Rearrange Figure 2 first and followed by Table 3. The authors have stated Figure 2 first in the text.

2. Line 280-286: The model for degradation has been finalized however, the explanation of the model itself is missing. For example, the degradation follows the Avrami-Erofeev equation with a high correlation coefficient, hence what’s the science behind this Avrami-Erofeev, why does Avarami-Erofeev mean in the degradation process? An explanation should be added to this.

Conclusion

1. Results of FTIR and kinetic are missing in the conclusion, at least the significant results to answer the objective of this manuscript.

Reviewer 3 Report

In this paper, the pyrolysis and co-combustion characteristics of semi-dry sewage sludge after the dehydration of flocculant and bituminous coal were studied. The paper can be accepted after considering the following recommendations.

(1) Some abbreviations should be explained such as SS, at the first time they appear in the main text. Authors should add a list of abbreviations

(2) The line thickness of the table frame should be uniform (Table 3), the font lines should be of uniform thickness.

(3) Some forms take up one page, some forms span the page, please layout appropriately.

(4) The research on the current situation in the introduction is not in-depth enough, and it is suggested to add several more related references.

(5) In the Introduction, the significance and practical application of the research should be more prominent.

(6) Please add a flowchart or photo of the equipment, so that the readers more clearly understand the experimental process.

(7) In the results and discussion, please add more analysis of reaction processes, such as changes in chemical reactions. Like from line 194 to 197, the authors only analyzed the changes of temperature, but did not analyze the reasons for the changes

(8) In Conclusion, the main points of each section of the article should be presented in a comprehensive and concise manner.

Round 2

Reviewer 1 Report

I think some questions haven't been answered properly. Authors should accept suggestions in order to improve the quality of the work and not be so inflexible. Especially in the case of retesting new samples with different temperature programs.